# How a co-design process led to more contextually relevant family planning interventions in emerging urban settings in Eastern Uganda

Jacquellyn Nambi Ssanyu[1,2,3]*, Rornald Muhumuza Kananura[1,2,4], Catherine Birabwa[1], Felix Kizito[5], Sarah Namutamba[5], Dorothy Akongo[5], Elizabeth Namara[5], Moses Kyangwa[5], Henry Kaula[6], Doreen Nakimuli[7], Andrew Magunda[6], Othman Kakaire[8], Peter Waiswa[1,2,4]

1 Department of Health Policy Planning and Management, School of Public Health, Makerere University College of Health Sciences, Kampala, Uganda, 2 Center of Excellence for Maternal, Newborn and Child Health, Makerere University School of Public Health, Kampala, Uganda, 3 Department of Women's and Children's Health, Uppsala University, Uppsala, Sweden, 4 Advance Innovations for Transforming Health in Africa, Kampala, Uganda, 5 Busoga Health Forum, Jinja, Uganda, 6 Kampala Slum Maternal and Newborn Health Project, Kampala Capital City Authority, Kampala, Uganda, 7 Population Services International, Kampala, Uganda, 8 Department of Obstetrics and Gynaecology, School of Medicine, Makerere university College of Health Sciences, Kampala, Uganda

* sanyukajacque@gmail.com

**Data Availability Statement:** The datasets used in the current study are available upon reasonable request. Data access requests can be submitted to

## Abstract

Voluntary, rights-based family planning upholds women's right to determine freely the number and spacing of their children. However, low-resource settings like Uganda still face a high unmet need for family planning. And, while urban areas are often indicated to have better access to health services, emerging evidence is revealing intra-urban socio-economic differentials in family planning utilization. To address the barriers to contraceptive use in these settings, understanding community-specific challenges and involving them in tailored intervention design is crucial. This paper describes the use of co-design, a human-centred design tool, to develop context-specific interventions that promote voluntary family planning in urban settings in Eastern Uganda. A five-stage co-design approach was used: 1) Empathize: primary data was collected to understand the problem and people involved, 2) Define: findings were shared with 56 participants in a three-day in-person co-design workshop, including community members, family planning service providers and leaders, 3) Ideate: workshop participants generated potential solutions, 4) Prototype: participants prioritized prototypes, and 5) Testing: user feedback was sought about the prototypes. A package of ten interventions was developed. Five interventions targeted demand-side barriers to family planning uptake, four targeted supply-side barriers, and one addressed leadership and governance barriers. Involving a diverse group of co-creators provided varied experiences and expertise to develop the interventions. Participants expressed satisfaction with their involvement in finding solutions to challenges in their communities. However, power imbalances and language barriers were identified by the participants as potential barriers to positive group dynamics and discussion quality. To address them, participants were separated into

the Makerere University School of Public Health Research and Ethics Committee at sphrecadmin@musph.ac.ug, with the subject, 'Request for access to Urban Thrive Project data'.

**Funding:** This study was supported by The John Templeton Foundation (Grant Number 62045 to PW and CB). The funder did not play any role in the design, implementation, analyses or reporting of these findings.

**Competing interests:** The authors have declared that no competing interests exist.

groups, and medical terminologies were simplified during brainstorming sessions. These changes improved participation and maximized the contributions of all participants. It is therefore important to consider participant characteristics and their potential impact on the process, especially when engaging diverse participant groups, and implement measures to mitigate their effects.

## Introduction

Voluntary, rights-based Family Planning (FP) upholds women's right to determine freely the number and spacing of their children and contributes to reduced maternal deaths, and improved well-being of women and their families [1,2]. However, communities in low resource settings continue to face challenges in accessing appropriate family planning options. In Uganda, the unmet need for FP remains high at 28% among married women and 32% among sexually active women [3]. Adolescent childbirths are also estimated at 54%, compounded with increasing repeat adolescent births at 56% and a 5.4% prevalence of short-birth intervals [4,5]. Contraceptive commodity stock-outs are also common, and a combination of provider, individual, and other health system challenges continue to affect the provision, access, and utilization of quality voluntary FP services [6].

Like other African countries, Uganda is undergoing increasing urbanization as shown in the recently named seven secondary cities [7]. However, while urban areas are often perceived and reported as having better health outcomes due to the wide availability of and apparent proximity to services [8], the urban communities are exposed to multidimensional factors that hinder them from accessing appropriate reproductive health services including FP. Emerging evidence is showcasing how intra-urban inequalities are increasing and driving poor health outcomes among the poor and other vulnerable populations in urban spaces [8,9]. For instance, lower levels of utilization of modern FP methods have been reported in slums in India [10], and a high unmet need for postpartum contraception was found among women in slum areas in Dakar city, Senegal [11]. Furthermore, research done in informal settlements in an urban municipality in central Uganda also showed that although the communities were well aware of FP and modern contraceptive use was relatively high at 42.7%, the unmet need for FP in these settlements was high, at 37.3%, higher than that reported nationally [12,13].

Several interventions have been shown to increase modern contraceptive uptake and reduce unintended pregnancies. High Impact Practices (HIPs) in FP are evidence-based practices that were identified based on their demonstrated magnitude of impact on contraceptive use and potential application in a wide range of settings [14]. High Impact Practices are classified as 1) enabling environment HIPs, those which address systemic barriers that affect the ability to access FP information and services; 2) service delivery HIPs, which improve the availability, accessibility, and quality of FP services and 3) social and behaviour change HIPs, which influence knowledge, beliefs, behaviours, and social norms associated with FP. Many of these HIPs have been applied in urban settings in Africa and elsewhere and shown to improve contraceptive behaviour with varying outcomes across poor and non-poor populations [15–17]. Lessons from implementing in urban areas highlight the need to tailor interventions to the specific needs of the different sub-populations in urban areas.

While existing National Reports and policies emphasize the urgent need for voluntary FP among adolescents and the inclusion of men in family planning programs [18,19] and the need to have specific interventions addressing urban communities' needs, there is a dearth of evidence on FP implementation best practices in urban settings. In this study, we apply the

Human-Centred Design (HCD) approach to actively engage communities in understanding their problems and collaboratively designing solutions.

The HCD approach can facilitate the refinement and adaptation of evidence-based interventions to the context of emerging urban spaces. Human-centred design refers to the use of techniques that communicate, interact, empathize and stimulate the people involved, obtaining an understanding of their needs, desires, and experiences resulting in products, systems, and services which are physically, perceptually, cognitively, and emotionally compatible with the full range of human characteristics including the basic and higher cognitive emotions [20]. The HCD approach is a process of ensuring that people's needs are met, that the resulting design is understandable and usable, and that it accomplishes the desired tasks [21]. The process ensures stakeholder engagement in the development of solutions and the design of programs through co-creation, inclusion, and transparency.

Co-design is an HCD tool used to simulate intuitions, opportunities, and possible futures for purposes of emersion, reflection, and discussion [20]. Sanders and Stappers [22] describe four interconnected phases of the co-design process: 1) pre-design research which aims to understand people's experiences in the context of their lives to prepare them to participate in codesigning; 2) generative research which aims to produce ideas, insights, and concepts that may then be designed and developed; 3) evaluative research which assesses the effect or effectiveness of the designed products, services or systems; and 4) post design research which focuses on how people experience the products or services. This process is iterative with the post-design phase leading into another design process. The terms co-creation and co-design are often used interchangeably but co-creation refers to any acts of collective creativity while co-design is a specific instance of co-creation which refers to the creativity of program designers and people not trained in design working together in the design development process [23].

Co-design has been applied in various fields, particularly, in designing digital health interventions such as in the development of a Mobile App to support informal caregivers in the United Kingdom to undertake regular exercise from home during and beyond COVID-19 restrictions [24] and to design a dietary intervention for adults at risk of Type 2 Diabetes in Australia [25]. Outside digital health, it has been used to develop a collective leadership intervention to improve the healthcare team safety culture in Ireland [26] and to develop a people-centred model of care for multi-morbid Chronic Obstructive Pulmonary Disease patients in rural Nepal [27]. There is however little evidence of the feasibility of using co-design in developing community-based interventions in low-resource urban settings.

There is a need for more examples of good practice and lessons from lower-income countries. To address this gap, this paper documents the use of the co-design approach to support the development of context-specific FP interventions in urban settings in Eastern Uganda.

## Methods

### Study setting

This work was done in December 2021 in Jinja city and Iganga municipality/town, located in Jinja and Iganga districts, respectively, in Busoga region, Eastern Uganda. Busoga region has one of highest fertility rates in Uganda, and according to Uganda's 2016 demographic survey, 21% of adolescents aged 15–19 years in Busoga region have begun childbearing; only 31% of married women are using any method of FP; 29% are using modern contraception; and 36% of married women have an unmet need for FP [3]. Jinja city is the largest secondary city in Uganda, while Iganga is an emerging town. Jinja city and Iganga municipality are located along the Uganda-Kenya transit route and host several commercial activities. The co-design process was done as part of a larger project, the Urban Thrive Project.

## The urban thrive project

The Urban Thrive Project is a three-year project being implemented in Jinja and Iganga in Eastern Uganda. The project is implemented by Makerere University School of Public Health, in collaboration with Busoga Health Forum, a Non-Governmental Organization working in Busoga region. The Project is aimed at improving coverage and voluntary uptake of FP by implementing interventions under three action areas: 1) ensuring communities are well knowledgeable about FP, 2) quality FP services are available and accessible, and 3) ensuring effective leadership. Several activities were proposed a priori at the start of the project under each of the three action areas (Table 1). More details about the Urban Thrive Project are presented in the study protocol [28].

## Objectives of the co-design process

Leveraging HCD principles, the co-design process sought to (1) identify barriers to the uptake of voluntary FP in urban settings and (any new) solutions to address these challenges; (2) to co-package high-impact interventions that are desirable, feasible, viable, and adaptable for these urban areas; and (3) to co-design implementation strategies to effectively deliver the selected interventions.

## Description of the urban thrive project co-design process

A five-stage co-design approach proposed by the Hasso-Plattner Institute of Design at Stanford [27,29] was applied to develop and test the intervention package. This is a non-linear, iterative process that provides a solution-based approach to problem-solving. This design process serves to understand the human needs involved, reframes the problem in human-centred ways, and adopts a hands-on approach to idea generation and testing. The five stages are: 1) Empathize, 2) Define, 3) Ideate, 4) Prototype, and 5) Test (Fig 1).

**Table 1. A priori proposed activities.**

| Activity | Description |
|---|---|
| **Result 1: Increased knowledge and understanding of voluntary FP among women, men, and young people.** | |
| Activity 1.1 | Training and supervision of gender and age-sensitive participatory community groups. |
| Activity 1.2 | Supporting and enhancing implementation of media-based social and behavioural change and use of digital technologies. |
| Activity 1.3 | Strengthening provider-initiated voluntary FP counselling. |
| **Result 2: Improved delivery of voluntary FP services.** | |
| Activity 2.1 | Developing/enhancing the knowledge and skills of healthcare providers. |
| Activity 2.2 | Strengthening the availability of FP commodities and services. |
| Activity 2.3 | Strengthening community-based provision of voluntary FP services. |
| Activity 2.4 | Strengthening the application and use of digital technologies to support service delivery. |
| Activity 2.5 | Improving referral care and management of FP side effects. |
| **Result 3: Governance and leadership** | |
| Activity 3.1 | Improving the alignment of FP services to reduce unmet need. |
| Activity 3.2 | Institutionalization and sustainability: Supporting better planning and integration of voluntary FP into urban plans |

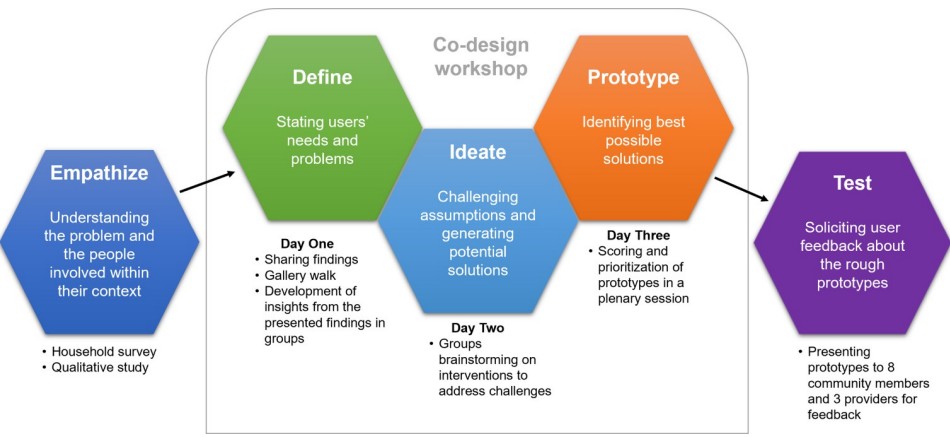

**Fig 1. The co-design process (Adapted from Plattner [29]).**

**1. Empathize.** This phase includes all the efforts to gain an empathic understanding of the problem and the people involved within their context. For this study, this was achieved through a concurrent mixed methods study involving a household survey and qualitative study, as part of the Urban Thrive Project's baseline assessment [28]. The household survey collected information on fertility (birth and pregnancy history), family planning use experience, sexuality (age at first sex, recent time of having sex, protected sex), knowledge of family planning methods, fertility, and family planning intention. A total of 1349 women aged 15–49 were enrolled into the study.

The qualitative study included adolescent girls and boys, men and women aged 15 to 54 years drawn from contextually diverse sites such as places of economic affluence, slums, and hard-to-reach island communities. District, city, and municipal health officials, community opinion leaders, and FP service providers were also interviewed as key informants. A total of 17 focus group discussions, 24 in-depth interviews, and 20 key informant interviews were done. Details of the data collection procedures are presented elsewhere [28].

**2. Define.** The Define stage involves stating users' needs and problems. It entails organizing all the information gathered during the Empathize stage to define the problem in a human-centred manner. Here, data collected from the survey and qualitative interviews were analysed by the research team and discussed into actionable problem statements of focus during ideation.

An in-person co-design workshop was then organized involving community members; political leaders at village and city/municipality level; religious leaders; FP service providers (from public, private-not-for-profit and private health facilities); representatives from district/city/municipal health departments; representatives from FP implementing partner organizations and media personalities. The participant categories were generated by the research team through a stakeholder analysis based on their influence on the uptake of FP, and their involvement in FP service provision and the governance of health services in the two sites. To ensure a diverse and inclusive representation, community members were selected to include an equal number of women and men from both Jinja and Iganga, including those from slum and non-slum settings as well as fishing communities. Furthermore, the selection process prioritized the inclusion of at least one adolescent participant from each site. These participants were reached through CHWs working with the project and who were familiar with the communities.

A total of 56 purposively selected individuals participated in the workshop, including 13 community members, 11 political leaders, 2 religious leaders, 14 FP providers, 9 health

authorities, 4 representative from partner organization working on FP, and 3 media personalities. The majority of the participants were male (37/56) and most (29/56) came from Jinja city. The community members comprised 5 men, 4 women (with one invited woman unable to attend), 1 adolescent boy, 1 adolescent girl, and 2 CHWs. The workshop was held over three consecutive days, each day lasting eight hours. It was facilitated by four independent facilitators who had training in and experience with the co-design process. These worked closely with the research and implementation teams. On the first day of the workshop, participants' expectations, hopes, and fears were assessed, and ground rules were established. Participants were then presented with the preliminary findings of the formative study and encouraged to develop insights from these findings. Insights reflect a deep understanding and knowledge of the problem that can inform and guide the design process. To achieve this, participants engaged in a gallery walk to study the findings and gain a deep understanding of the drivers underlying the identified FP issues. Preliminary findings from the formative study that were presented to the participants are summarised in Table 2. The quantitative findings are also published elsewhere [30].

Participants were divided into three groups, leaders, FP service providers, and community members, to develop insights about different FP issues and ascertain which FP challenges were most important to them. The insights were organized into three: 1) those that related to the demand side, 2) those that related to the supply side, and 3) those that related to leadership and governance. Finally, participants scored the insights generated for each of the three action areas.

**3. Ideate.**   The ideation phase involves challenging assumptions and the generation of potential solutions to address identified problems. On the second day, the focus was on understanding and applying the design approach. The facilitators introduced the participants to the co-design approach. The participants, in their initial groups (community members, leaders, and service providers), were guided through brainstorming sessions on interventions to address the challenges identified. Focus was on three design challenges, one derived from each of the three project action areas:

1.  How might we increase knowledge and understanding of voluntary FP among women, men, and young people in towns and cities?

2.  How might we ensure consistent provision of quality voluntary FP services in emergent towns and cities?

3.  How might leaders build the capacity to lead or manage FP programs or create an enabling environment to improve organizational and management systems for FP provision?

Each participant group discussed interventions to address all three design challenges.

**4. Prototype.**   This phase aims to identify the best possible solution for each of the problems identified during the first three stages. On the third day, the participants in a plenary session scored and prioritized the prototypes that had been developed in the brainstorming sessions on Day Two.

**5. Test.**   This stage involved taking the rough prototypes to the target communities for feedback. Different components and features of the rough prototypes were presented and explained to 11 individuals who had not participated in the co-design workshop for their opinions and suggestions. The facilitation team, in collaboration with CHWs, employed purposive sampling to select the participants. The sample consisted of 8 community members (1 adolescent girl, 2 women, and 5 men) and 3 FP service providers. The goal was to understand the end user view on each prototype, in terms of its fit for the target populations and communities. This was done by simulating a real-life scenario, so as to tweak components of each prototype

**Table 2. Key action areas and findings from the formative study.**

| Action area | Key findings |
|---|---|
| *Family planning use* | |
| Modern FP use and the individual and socio-economic factors associated | ■ Overall modern FP use was estimated at 48%.<br>■ Modern FP use reduced with increase in age.<br>■ Increase in parity was positively associated with FP use.<br>■ Compared to mainland non-slum and landing site slum residents, mainland slum residents were two-fold and-three-fold more likely to use modern FP, respectively.<br>■ The odds of modern FP use increased with wealth status, but wealth mattered only in non-slum mainland areas. Only women in poor households in non-slum mainland areas were less likely to use FP than those from households of higher wealth status.<br>■ Women with secondary or higher level of education were less likely to use FP than those of lower education.<br>■ Women whose decisions to use FP were independent of other authorities were more likely to use FP.<br>■ Regarding religion, overall, the likelihood of using modern FP was lowest among Muslim and Catholics, as compared to Pentecostals and Anglicans [30]. |
| *Demand side challenges* | |
| Knowledge of FP and fertility | ■ Whereas awareness about male condoms as a method of FP was high, that of methods like injectables and the Standard Days Method was low. |
| Barriers to contraceptive use | ■ Several women reported experiencing contraceptive-related side effects such as bleeding irregularities, loss of sexual desire and constant headaches.<br>■ Myths and misconceptions about FP were also common.<br>■ Economic hardships limited access to FP and reduced young women's decision-making power for contraceptive use. |
| FP decision making | ■ Some women reported spouse control over decision-making for contraceptive use while joint-decision making was also still low. |
| *Supply-side challenges* | |
| Commodity stock-outs | ■ Rampant stock-outs of reproductive health commodities and supplies at health facilities, including gloves and syringes. |
| Contraceptive counselling | ■ Health workers did not provide adequate information about the side effects of FP. |
| Integration of services | ■ Lack of integration of FP services with other health services. |
| *Leadership and governance challenges* | |
| Coordination of non-government health partners | ■ Challenges coordinating implementing partners in terms of resource mobilization and reporting because of parallel systems. |
| Adolescent sexual reproductive health policies | ■ Unclear policies about adolescent sexual reproductive health education and provision of FP service. |

and refine them for improved delivery. Feedback was sought on community and service provision prototypes only.

## Prioritization of interventions to implement

Findings from the co-design workshop and the prototype testing were discussed in a project team meeting, involving the research team from Makerere University School of Public Health and the implementation team from Busoga Health Forum. The meeting was aimed at prioritizing interventions to implement based on the identified needs of the target populations and the availability of resources. Some modifications were made to the proposed interventions: some similar interventions were merged while the implementation strategies of others were clarified.

### Ethical considerations

The Urban Thrive Project was approved by Makerere University School of Public Health Research and Ethics Committee (Reference Number: SPH-2021-146) and Uganda National Council for Science and Technology (UNCST) (Reference Number: HS1826ES). Administrative clearance was also obtained from the leaders of Jinja city and Iganga municipality. Written informed consent was sought from all participants involved in the formative research and the co-design workshop. For minors (excluding emancipated minors), consent to participate in the formative research and the co-design workshop was sought from the household heads. And, whereas the study team had access to information that could identify individual participants, none of this was linked to the data used or presented in this work. All data files were stored on password-protected computers to ensure privacy and confidentiality.

## Results

### Co-design process

**Attendance and participation.** The workshop had a satisfactory attendance and participation rate across the three days. Specifically, 82% of participants (46/56) attended all three days, while 93% (52/56) attended two days. Four attendees left after the first day, including a political leader, two media persons, and an implementing partner, citing personal reasons and other commitments. The diverse group of co-creators provided a wide range of experiences, ideas, and expertise, which were critical in developing the prototypes. The participants, particularly community members, were satisfied with the process's structure and their involvement in devising solutions to their community's challenges.

**Group dynamics.** Participants expressed three concerns on the first day of the workshop which they believed would affect the quality of discussions, especially, between members of the community and *"the experts*, *like doctors"*. These included the use of complicated medical terminology, the power differences between the community, leaders, and providers, and the use of English as the primary language during discussions. To mitigate these concerns, efforts were made to simplify medical terminology and incorporate local languages into the discussions. At the end of the workshop, participants reported that the discussions were comprehensible. To ensure equitable participation and mitigate power imbalances, participants were grouped into three categories: FP providers, community members, and leaders. This enabled, especially, the community members to discuss comfortably among themselves and have one representative report back in the plenary session. Additionally, targeted engagement with each stakeholder group maximized their individual contributions.

While we observed no substantial differences in participation levels between men and women both in the group discussions and plenary sessions, we did notice that adolescents were relatively less expressive, particularly on the first day. However, their engagement noticeably improved in the subsequent days.

**Participant reactions to the formative study findings.** Overall, the participants agreed with the findings presented as an accurate representation of FP utilization in their communities. Religious leaders, notably the Muslims, provided insightful comments explaining the observed differences in FP usage, attributing them to religious beliefs opposing hormonal methods due to their associated side effects. They also mentioned a preference for withdrawal and natural methods, even though health workers did not provide adequate information about them. Additionally, participants expressed surprise at the higher FP usage among mainland slum residents compared to those from non-slum settings, speculating that this could be due to the presence of marginalized poor populations in these areas, such as domestic workers.

**Table 3. Key insights prioritised under each action area.**

| Focus area | Key insights, in order of importance |
| --- | --- |
| Demand side | 1) There are negative attitudes and perceptions of clients towards FP |
| | 2) Side effects are a major deterrent to FP uptake |
| | 3) There is a need to provide more information or facts about FP |
| | 4) Myths and misconceptions about FP are a big barrier to the uptake of voluntary FP |
| | 5) Men are key decision-makers, but they are not involved in FP programming |
| | 6) Poverty makes adolescents vulnerable to unsafe sexual behaviour |
| Supply-side | 1) There are insufficient quantities of FP commodities, and some facilities experience stockouts of FP commodities and supplies |
| | 2) There is inadequate counselling of clients before FP initiation, especially about the side effects |
| | 3) Inadequate investigations are done by providers to determine eligibility |
| | 4) Providers who are not primarily responsible for the provision of FP in health facilities are reluctant to provide FP |
| | 5) Biased FP provision towards some FP methods based on donor or partner priorities |
| | 6) Negative attitudes towards FP by the community including key opinion leaders like the religious leaders. |
| Leadership and governance | 1) There is poor coordination between departments in promoting FP (within health and across, such as education and community health services) |
| | 2) Unfavourable policies to promote the use of FP by adolescents in schools |
| | 3) There is limited human resource in public health facilities to support FP service provision. |
| | 4) There are limited resources/funds allocated to FP at the city/municipality level |
| | 5) Insufficient supplies in health facilities |
| | 6) Negative attitudes towards FP by the community including the key opinion leaders |

**Model development.** In the Define Stage, six insights were prioritised under each area of focus (demand, supply, and governance) (Table 3).

To address the challenges identified, fourteen prototypes were developed during the co-design workshop: eight prototypes were targeting demand-side challenges; four targeted supply-side challenges and two targeted governance (Table 4). Details of each prototype are also presented in S1 Table.

The community members and service providers involved in the prototype testing overall concurred with the features of the prototypes generated from the co-design process. A few modifications were suggested, and these were incorporated into the features of the prototypes earlier developed. An additional prototype, 'Facility-led follow-up mechanism for users of FP', was generated by FP service providers during prototype testing to address the supply-side barriers. Feedback during testing included:

1. **Demand side Prototype 1 (*'Izima'* or *'Amazima ku famire planning'* campaign):** For clarity and ease of comprehension by the target audience what the campaign is about, the participants recommended that the campaign is called '*Amazima ku famire planning* campaign', translating to, 'the truth about family planning' and not *'Izima'*, which translates to 'the truth', as earlier proposed in the co-design workshop.

2. **Demand side Prototype 2 (Target parents to reach the youths and adolescents) and Prototype 3 (Target men and provide facts and information about FP):** They maintained that parents and men, particularly, should be found at their workplaces and not at home. They noted that some parents do not attend parents meetings at school. Their supervisors at the workplaces could be used to mobilise them during lunchtime or very early in the

**Table 4. Prototypes developed to address the demand, supply, and governance challenges.**

| Design Challenge | Prototypes |
|---|---|
| 1) How might we increase knowledge and understanding of voluntary FP among women, men, and young people in towns and cities? | Prototype 1: *'Izima'* or *'Amazima kufamire planning'* [the truth about family planning] campaign |
| | Prototype 2: Target parents to reach the youths and adolescents |
| | Prototype 3: Target men and provide facts and information about FP |
| | Prototype 4: *'Beeda or situla Kyosobola'* [carry what you can manage] campaign |
| | Prototype 5: Community groups to promote uptake of FP in towns/urban centres |
| | Prototype 6: Use social media and digital technologies to promote behaviour change and uptake of VFP |
| | Prototype 7: Use Interpersonal (peer-to-peer) communication to promote FP uptake |
| | Prototype 8: Provider-led counselling to promote uptake of voluntary FP |
| 2. How might we ensure consistent provision of quality voluntary FP services in emergent towns and cities? | Prototype 1: Enhance the knowledge and skills of providers to provide quality FP services through training and mentorships |
| | Prototype 2: Ensure affordable FP commodities are available in private clinics and at the community level |
| | Prototype 3: Create a platform for providers to learn and exchange ideas and information about FP |
| | Prototype 4: Timely management and referral for side effects management |
| 3. How might leaders build the capacity to lead or manage FP programs or create an enabling environment to improve organizational and management systems for FP provision? | Prototype 1: Increase knowledge and awareness about FP among leaders: *'Fuuka eyebuzibwaaku kubya famire planning'* [Become someone who can be consulted about family planning] campaign. |
| | Prototype 2: Orient technical and political leaders in planning, coordination and monitoring of FP activities in their areas. |

morning depending on when they are less busy. For instance, mechanics are less busy in the morning hours, and may be easier to mobilise during that time. In addition, men could also be targeted at sports events and during sports programmes on radio and television.

3. **Supply-side Prototype 5 (Facility-led follow-up mechanism for users of FP):** This prototype was generated while soliciting feedback on other prototypes. The providers proposed that each FP clinic have a database of clients who have received FP. Clinic staff would then make follow-up phone calls to clients and support them in side effect management. They would also send clients text message reminders, messages with facts about FP, information about side effects and what clients need to do in case they experience any side effects.

## Final model developed

Out of the 14 prototypes initially developed during the co-design workshop and 1 developed during prototype testing, 10 interventions were finalised and prioritised for implementation, including 5 targeting the demand-side barriers to FP uptake, 4 targeting the supply-side barriers and 1 targeting leadership and governance. Some of the interventions came out of modifications of the interventions specified a priori at the beginning of the project, while others were identified during the workshop. A comparison of the activities proposed a priori, prototypes

**Table 5. Activities proposed a priori, prototypes developed during the co-design workshop and final interventions developed.**

| A priori specified activities | Matching prototypes developed during the co-design workshop | Final intervention developed |
|---|---|---|
| **Demand side** | | |
| Training and supervision of gender and age-sensitive community groups | Use of community groups to promote uptake of voluntary FP | Community group engagement |
| Supporting and enhancing implementation of media-based social and behavioural change and use of digital technologies. | 'Izima' or 'Amazima kufamire planning' [the truth about family planning] campaign to disseminate FP information through local leaders, CHWs, radio talk shows, drama shows, social media, community champions and using information, education and communication material | Use of radio talk shows on public and community radios to disseminate FP information embedded in the *Amazima kufamire planning* campaign |
| | | Drama group performances to disseminate FP information embedded in the *Amazima kufamire planning* campaign |
| | Use social media and digital technologies to promote behaviour change and FP uptake | Use of SMS and social media to disseminate FP information |
| Strengthening provider-initiated voluntary FP counselling. | Provider-led counselling to promote uptake of voluntary FP | Health worker training and supportive supervision |
| | *'Beeda or situla Kyosobola'* [carry what you can manage] campaign | *'Beeda Kyosobola'* [carry what you can manage] campaign |
| | Use Interpersonal (peer-to-peer) communication to promote FP uptake | |
| | Target parents to reach the youths and adolescents | |
| | Target men and provide facts and information about FP | |
| **Supply side** | | |
| Developing/enhancing the knowledge and skills of healthcare providers | Enhance the knowledge and skills of providers to provide quality FP services through training and mentorships | Health worker training and supportive supervision |
| Improving referral care and management of FP side effects. | Timely management and referral for side effect management | |
| | Facility-led follow-up mechanism for users of FP | |
| Strengthening the availability of FP commodities and services. | | Commodity manager training and supportive supervision |
| Strengthening community-based provision of voluntary FP services. | Ensure affordable FP commodities are available in private clinics and at the community level | Community health worker training and supportive supervision |
| Strengthening the application and use of digital technologies to support service delivery. | Create a platform for providers to learn and exchange ideas and information about FP | Use of social media groups to support service delivery |
| **Governance and leadership** | | |
| Improving the alignment of FP services to reduce unmet need. | Orient technical and political leaders in planning, coordination and monitoring of FP activities in their areas. | Quarterly review meetings to support better planning and integration of FP into urban plans and improve alignment of FP services to meet the need. |
| Institutionalization and sustainability: Support better planning and integration of voluntary FP into urban plans | Increase knowledge and awareness about FP among leaders: *'Fuuka eyebuzibwaaku kubyafamire planning'* [Become someone who can be consulted about family planning] campaign. | |

generated during the co-design workshop, and final interventions developed is provided in Table 5.

The final prototype model prioritized for implementation is illustrated in Fig 2. Details of the specific intervention components and associated delivery approaches are also described below.

## Demand-side focused interventions

### 1. Community Group Engagement

Twenty locally generated social groups will be engaged to deliver targeted information on FP to address key knowledge gaps, norms and other concerns that deter contraception use and

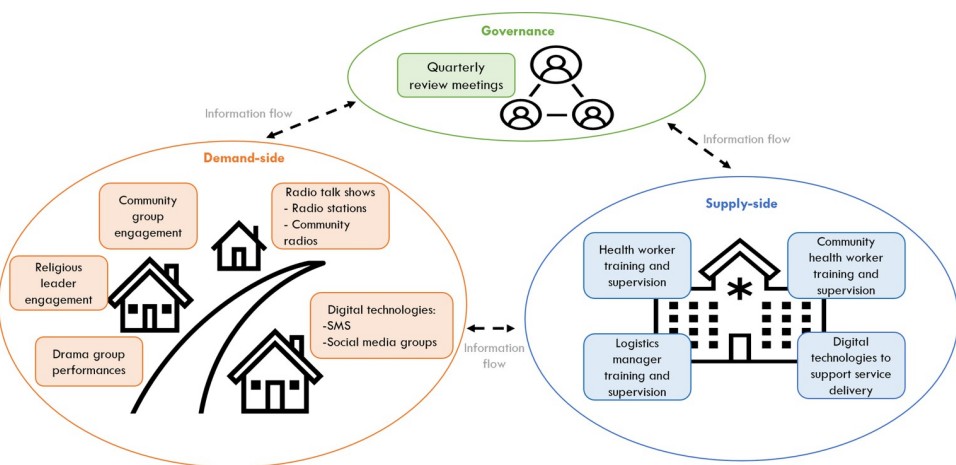

**Fig 2. The intervention model prioritized for implementation.**

promote healthy reproductive behaviours. Whereas this intervention did not directly emerge out of the brainstorming sessions, but was rather presented to the participants to deliberate upon, it was prioritized for implementation given the strength of evidence for it as a high-impact practice in FP [14]. The co-design process helped refine its components and delivery strategy.

Ten facilitators will be recruited and trained to mobilize group members and facilitate group discussions. Women, men, and adolescents will be targeted through their work niches, saving groups and other dominant social groups or community platforms. Facilitated participatory learning and action (PLA) cycles [31] will be used for community mobilization. This is a four-phase process in which the groups collectively decide priority actions, guided by a trained facilitator: Phase 1: the groups identify and prioritise FP issues or perceived local barriers; Phase 2: the groups identify feasible strategies to address those issues; Phase 3: The groups implement their strategies; and Phase 4: the groups assess their performance. Facilitators will conduct at least one meeting every month, but in the first three months, the groups will meet twice a month. The participating community groups will also be expected to hold community meetings to raise awareness of key FP problems, discuss their proposed strategies and provide feedback to the communities on the actions and progress.

From the co-design workshops, the '*Amazima ku famire planning*' campaign will be rolled out in the groups, emphasizing providing 'the truth about FP'— sharing complete information, including information about the side effects and how to manage them. The PLA cycles and FP information dissemination will be embedded within the routine group meeting schedules, and no new independent meeting days will be created for the purpose. Each group will also be attached to a trained CHW for facilitator mentorship and supportive supervision.

**2. Radio talk shows**

*Talk shows on public radio stations*: The project will facilitate the airing of tailored radio messages through two popular or preferred radio stations in the region, one targeting listeners in Jinja city and the other listeners in Iganga municipality. The information shared will be based on the '*Amazima ku famire planning*' campaign aimed at dispelling myths about FP and how to manage side effects. Trained health workers will be used to facilitate the talk shows. Community champions, or 'happy users' will also be used to disseminate correct information about FP.

*Talk shows on community radios*: The project will also use CHWs who have received training about FP to conduct talk shows sharing information about FP on outdoor community

radios located in marketplaces to target youth and parents that work in the trading centres. This is also aimed at reaching community members that may not have radios at home.

### 3. Use of digital technologies to disseminate FP information

*Short Message Service*: Short Message Services (SMS) will be used to share weekly tailored FP messages to mobile phone subscribers. Tailored messages will be disseminated to different sub-populations, ensuring fair representation of older women, older men, adolescents, and young adults. Participants shall be identified by CHWs from community members within their areas of operation. Written informed consent shall be obtained from all those that choose to participate in the program. Their details shall be captured using a profiling tool that will collect information including their sex, age, contraceptive use status, and language of preference. These will then be used to tailor messages to the recipients. All messages shared shall also be aligned with the '*Amazima ku famire planning'* campaign.

*Social media platforms*: Social media platforms, particularly, Whatsapp, will be used to increase access to FP information and linkage to other resources to target, especially, the youth and men, who have been shown to have higher ownership of smartphones. About 10 groups each of 50 participants will be created per year. Influential individuals will be identified who will act as group administrators to oversee overall group operations. Administrators will also be responsible for identifying and inviting eligible members to join the group. As an adaptation from the co-design workshop, each group will have at least one trained FP provider who will disseminate tailored FP messages and also respond to any issues raised by the group members. Messages will also be based on the '*Amazima ku famire planning'* campaign.

### 4. Drama Groups

The project will engage one popular local drama groups in each city/municipality to present skits on FP in their communities targeting schools and informal workplace settings grounded in the '*Amazima ku famire planning'* campaign. These groups shall be expected to perform twice a month for at least 24 months, expecting at least 20 people in the audience. The groups will be expected to draft scripts for their drama performances based on topics identified by the project team. The scripts shall be reviewed by the team and then developed into plays. A new script shall be prepared every six months. At least one trained CHW shall be in attendance at every performance to respond to technical questions raised by the audience.

### 5. *'Beeda kyosobola'* [carry what you can manage] campaign

This campaign was proposed in the co-design workshop as a way to engage religious leaders in churches and mosques to disseminate FP information. The leaders will be engaged and provided with FP information to share on their platforms. The focus for the campaign shall primarily be on educating about child spacing and having manageable family sizes and not necessarily having few children. The campaign shall also be used by religious leaders to sensitize men about the advantages of having families they can ably provide for.

## Supply-side focused interventions

### 1. Health worker training and supportive supervision

Working with the city reproductive health section and using data from a health facility assessment, the project team will establish training gaps for FP providers and prioritize categories of providers and places that have the greatest capacity gaps for the provision of contraceptive methods. The project then aims to train 120 providers, including nurses, midwives, clinical officers, medical officers, and drug shop and pharmacy staff. A mixed-method training

approach will be adopted including classroom training, on-job mentorship, and simulation-based learning techniques. The providers will be trained to enhance their knowledge, skills, and competencies in the provision of quality and person-centred FP services and to manage contraceptive-related side effects. The trainings will be facilitated by expert trainers and peer mentors using existing national FP training guidelines and manuals.

Although provider training was proposed as one of the project activities at the start of the project, several adaptations were made during the co-design workshop. Participants recommended identifying more than one staff in each health facility to be trained to ensure continuity even if one of those trained is unavailable. In bigger health facilities, they also proposed training providers that do not routinely provide FP, for instance, those in outpatient departments and immunization sections to promote the integration of services. Trained staff could cascade skills to colleagues through Continuous Medical Education sessions (CMEs).

The trained providers will receive quarterly supportive supervision and mentorship focused on strengthening the correct assessment of client FP needs and eligibility, client-centred method prescription, integrated service provision, respectful care, and youth-friendly care provision among others. From the co-design workshop, participants recommended forming provider committees for quality improvement at the city or municipal level, mandated with quarterly support supervision, CMEs and provider mentorships.

### 2. Community health worker training and supportive supervision

Thirty CHWs will receive training with an emphasis on the provision of adolescent-friendly contraceptive services and effective male engagement. This will be done to strengthen their capacity to correctly assess clients for FP, make appropriate method recommendations, disseminate gender-sensitive or age-appropriate information on FP or reproductive health, refer and follow-up of clients and identify and manage side effects among other basic FP services. The project will also support quarterly CHW supervision by FP focal persons in each site to promote skill retention and support the CHWs in identifying solutions to the challenges hindering their optimal performance. During the supervision, the utilization of information generated by the CHWs through their reports by the health facilities will be reinforced. The project will also support the integration of CHWs by improving linkages between service delivery points and CHWs as well as households/families.

As an adaptation from the co-design process, the trained CHWs will be provided with information, education and communication materials developed based on the '*Amazima ku famire planning*' campaign. To support their work and for motivation, they will also be provided with modest transport facilitation for activities and also given reporting tools. And to improve linkage to health facilities, each CHW will be attached to a public health facility for referral in case of any side effects, easy supervision of their activities, and for provision of commodities and supplies.

### 3. Commodity manager training and supportive supervision

The project will train Supply Chain Management (SCM) officers at the sub-national level and the facility managers/store managers from about 10 facilities in each site to enhance their capacity to manage FP commodities. The trained logistics managers will receive quarterly on-job mentorship to improve SCM performance in the areas of organization, processes, technical infrastructure, data capture and management. This is intended to increase data visibility and use. This intervention was not proposed by the participants during the workshop but was prioritised for implementation given the participants' ranking of commodity stock-outs as a supply-side barrier to the uptake of voluntary FP.

**4. Use of social media groups to support service delivery**

Based on Prototype 3 of the supply-side focused interventions, Whatsapp groups of trained health workers, health facility FP focal persons and commodity managers shall be created in each site to facilitate the timely exchange of ideas and information on stock levels and availability of products in different health facilities, for client referrals for side effect management and in case of stock-outs and for exchange of health information among the health workers. This was a new intervention proposed during the workshop that was prioritized for implementation owing to its relevance and expected low cost of implementation.

**Governance-focused interventions.**   *Quarterly review meetings*. The project will support or develop systems to collate and analyse relevant data to support decision-making by cities/towns relating to estimating supply needs or commodity distribution, as well as for process and performance improvement. This evidence will be shared with the city/town leaders and other key actors in quarterly review meetings to determine facilities or areas that are priorities for action and to inform decisions on improving the organization and management of FP services. The meetings shall also be used to educate the urban authorities about FP to enable them disseminate it to their communities by virtue of their positions and access. Through the meetings, the leaders will also be supported to develop evidence-based plans for FP services to ensure that services are where the clients are and minimise missed opportunities.

## Discussion

This paper describes the process and lessons learnt from using co-design, an HCD tool, to understand barriers to the uptake of voluntary FP, identify new solutions to increase the uptake of voluntary FP in urban settings; to co-package high-impact interventions that are desirable, feasible, viable and adaptable for these urban areas; and to co-design implementation strategies to effectively deliver the selected interventions in urban settings in Eastern Uganda. The co-design process described facilitated the development of a package of interventions to address demand-side, supply-side, and leadership and governance barriers to uptake of FP in these urban settings through, first, the adaptation of high-impact interventions initially specified for implementation at the start of the project and by generating new context-specific interventions. During this process, power imbalances and language barriers were identified as key issues that could affect the group dynamics and discussion quality. To address these challenges, participants were separated into groups, and medical terminologies were simplified which improved participation and maximized the contributions of all participant groups.

One of the key outcomes of the process was the development of the '*Amazima ku famire planning*' or 'the truth about family planning' campaign. Its aim was to address information gaps in FP client counselling, particularly about the side effects of FP and how to manage them. This campaign was integrated into all project social behaviour change communication interventions, including radio talk shows, social media and SMS messaging, and drama shows. Concerns about the side effects of contraceptives have been cited as one of the main barriers to the utilization of services by women, men, and young people both in Uganda [32–34] and elsewhere [35,36]. Contraceptive-related side effects not only inhibit the initial uptake of FP but are also lead to method switching and discontinuation [37]. And, whereas, providing complete information about the side effects could better prepare women to cope with them and reduce the spread of misinformation, health workers often do not disclose complete information about contraceptive-related side effects and how to manage them. According to a national-wide survey, 4 out of 10 contraceptive users in Uganda were not told by their providers about potential side effects they may encounter when they were initiated on FP [38]. This campaign, therefore, has the potential to impact FP uptake and continuation rates positively. However, it

is crucial to monitor for any adverse effects of the campaign, such as propagating the spread of misinformation and increasing attitudinal resistance.

Whereas some of the interventions prioritized for implementation after the co-design process were specified a priori, adaptations were made to them to increase acceptance and adoption by the target populations. Community group engagement, for instance, is a promising social and behaviour change high-impact practice in FP for which good evidence exists but more research is needed to fully document implementation experience and impact [14]. It has been shown to be a cost-effective way to improve preventive and care-seeking behaviours that lead to reducing neonatal mortality [39,40]. There is therefore need for more evidence of its effectiveness in improving contraceptive behaviour, more so in urban settings. As an adaptation to increase the adoption of the intervention, participants in the co-design workshop recommended plugging FP education sessions into routine group meetings and attaching a trained CHW to each group. The recommendation to use routine meeting days is especially relevant in urban settings like Jinja city and Iganga town where a lot of people are involved in income-generating activities that require more time commitment and may not have as much time to attend meetings. Additionally, attaching a trained CHW is important for the supervision and ongoing mentorship of the trained facilitators.

Integrating trained, equipped and supported CHWs is also a proven service delivery high-impact practice [14] that was prioritized for implementation with recommendations to take into consideration CHW motivation and support. CHW activities in Uganda are voluntary and largely unpaid which leaves the CHWs with the dilemma of balancing the provision of quality health services with limited support [41]. As such, for the success of the CHW activities, particularly in this urban context where people's livelihoods are tied to daily income, CHW motivation is a key aspect for consideration. Relatedly, in line with the high-impact practice to integrate FP services with other health services [14], the participants recommended training health workers from departments that do not primarily provide FP services to promote integration and training more than one health worker at each facility for continuity of services, which were incorporated into the final implementation model.

Some new interventions also came out of the workshop. Health worker social media groups to facilitate information sharing and client referral were adopted, as a less resource-intense intervention. This intervention holds particular relevance in urban settings where access to mobile phones and the internet is more widespread. Likewise, the *'Beeda kyoosobola'* or 'carry what you can manage' campaign, aimed at engaging religious leaders to encourage manageable family sizes was adopted from the workshop. Bormet, Kishoyian (42) from their work in Kenya and Zambia working with Faith-Based Organizations and religious leaders maintained that religious leaders and faith-based organizations can be strong and trusted advocates for FP. They recommended attention to terminology around FP to avoid misunderstanding FP [42], similar to suggestions from our co-design workshop where participants recommended emphasis be placed on child spacing and having manageable family sizes, and not restricting the number of children.

The adoption of a co-design methodology facilitated the integration of key stakeholders in FP programming to formulate strategies aimed at improving the reach and acceptance of interventions to increase the uptake of voluntary FP. The direct experience of FP users with the challenges in accessing and utilising FP services informed the appropriateness of demand-side and service delivery interventions developed, relative to their specific context and capacity. Similarly, the engagement of FP service providers and managers ensured that interventions were customized to their requirements, exploiting existing opportunities, such as a vibrant private sector to increase access to services and social media for information exchange, while recognizing the gaps and limitations within the health system they operate. Furthermore, co-

designing interventions with the leaders and beneficiaries of the interventions has the potential to enhance the institutionalization and sustainability of the interventions [43].

The co-design process revealed the significance of participant characteristics and their potential to influence group dynamics and the need to put in place measures to minimise the effect of this on the outcomes. In this project, the power dynamics between health workers, leaders, and the community could have impacted discussions; however, the division of participants into groups addressed this issue. Pallesen, Rogers [44] while using co-design to develop a collective leadership education intervention for healthcare teams also reported that many participants expressed apprehension about their ability to contribute to a process that is unfamiliar while some felt that other team members were more highly qualified than themselves. They established a positive team climate by holding discussions in small groups, through collective leadership, and by allowing participants to influence the direction of the workshops through open discussions similar to how our process was designed. Ní Shé and Harrison [45] also emphasised that reflective discussions and conscious efforts to eliminate power imbalances are crucial for inclusive and equitable involvement in the co-design process to enhance health outcomes.

Despite the strengths of the co-design process in generating innovative context-specific interventions to increase the coverage and uptake of voluntary FP services, this process yielded several interventions for which little evidence of effectiveness or sustainability exists and some of which could not be implemented by the project given the time and financial resources available. These have been presented in this paper as potential interventions for consideration in future FP programs in similar emergent urban settings to allow the generation of evidence of feasibility, effectiveness, and sustainability.

Another limitation of this study was the lack of gender balance in the participants included in the co-design workshop. Although the study team made efforts to ensure equal participation of men and women in the process by assigning quotas to male and female participants in each participant category, it was easier to implement for community members but harder for other participant groups, like the leaders, urban health authorities and opinion leaders. This was because invitations to these participants were made based on their positions and, due to systemic gender bias, these positions are predominantly occupied by men. Consequently, valuable feminine perspectives that could have enriched the design of governance-focused interventions may have been missed. Future projects should be cognisant of this lack of representation of women in positions of power and influence and make more deliberate efforts to ensure gender-equitable participation at all levels of the co-design process.

Relatedly, in this project, the participation of adolescents in the discussions was notably limited, even within the community participant group. This could be attributed to the fact that there were only two adolescents present, leading them to possibly feel hesitant in expressing their thoughts and opinions. Moreover, they may have experienced a sense of discomfort or intimidation when engaging in discussions with the adults, given the profound power imbalance prevalent in the relationships between adults and children in Uganda [46]. In order to foster more meaningful engagement of adolescents in addressing their barriers to FP uptake, deliberate measures should be taken to avoid tokenism [47]. First, it is important to ensure equal representation of adolescents alongside adults, enabling their voices to be heard and valued. Additionally, seeking their perspectives on the methods of engagement is vital to ensure their effective participation. This includes considering whether they prefer to participate as a group of adolescents alone or alongside the adults.

Lastly, it is important to acknowledge that this study has limitations due to the small number of participants included per participant group. This limited sample size may not have provided sufficient representation of all urban sub-populations, taking into account factors such

as age, socio-economic status, religion, or ethnicity. Consequently, while the study aimed to generate context-specific interventions, there is a possibility that the interventions generated may not fully address the diverse needs of all populations.

## Conclusion

The co-design process facilitated the development of a package of interventions to address demand-side, supply-side and leadership and governance barriers to the uptake of voluntary FP in these urban settings through the adaptation of high-impact practices in FP and the generation of new context-specific interventions. For the success of the process, it is important to pay attention to participant characteristics, including the differences in age, education levels and position in the community, and how these could influence group dynamics and the quality of the discussions and to put in place measures to minimise the effects of this on the process. The intervention model prioritized for implementation will undergo further adaptations, in line with the human-centred design process, and rigorous testing to assess its effect on the uptake of voluntary FP and the reduction of unmet need for FP in emergent urban settings in Eastern Uganda. Findings from the outcome evaluation, describing what worked, and the process evaluation, detailing how the context and implementation affected the effectiveness of the interventions will also be published to share learnings.

## Supporting information

**S1 Table. S1 Table presents comprehensive details on the fourteen prototypes developed collaboratively by the participants during the co-design workshop.** These prototypes were aimed at addressing challenges on the demand-side, supply-side, and governance aspects. Out of the fourteen prototypes, eight were designed to target demand-side challenges, four to tackle supply-side challenges, and the remaining two prototypes focused on governance challenges. (DOCX)

## Acknowledgments

We thank the city/town leadership in Jinja and Iganga for their support during the co-design process. We also thank the community members, service providers and urban health authorities that participated in the co-design workshop. Lastly, we acknowledge the contribution of the team that facilitated the co-design workshop for their input and guidance throughout the process.

## Author Contributions

**Conceptualization:** Jacquellyn Nambi Ssanyu, Rornald Muhumuza Kananura, Catherine Birabwa, Othman Kakaire, Peter Waiswa.

**Data curation:** Jacquellyn Nambi Ssanyu, Rornald Muhumuza Kananura.

**Formal analysis:** Jacquellyn Nambi Ssanyu.

**Funding acquisition:** Rornald Muhumuza Kananura, Catherine Birabwa, Othman Kakaire, Peter Waiswa.

**Investigation:** Jacquellyn Nambi Ssanyu, Rornald Muhumuza Kananura, Catherine Birabwa, Othman Kakaire, Peter Waiswa.

**Methodology:** Jacquellyn Nambi Ssanyu, Rornald Muhumuza Kananura, Catherine Birabwa, Felix Kizito, Sarah Namutamba, Dorothy Akongo, Elizabeth Namara, Moses Kyangwa, Henry Kaula, Doreen Nakimuli, Andrew Magunda, Othman Kakaire, Peter Waiswa.

**Project administration:** Jacquellyn Nambi Ssanyu, Felix Kizito, Sarah Namutamba, Dorothy Akongo, Moses Kyangwa, Peter Waiswa.

**Resources:** Jacquellyn Nambi Ssanyu, Catherine Birabwa, Moses Kyangwa, Peter Waiswa.

**Software:** Jacquellyn Nambi Ssanyu.

**Supervision:** Jacquellyn Nambi Ssanyu, Rornald Muhumuza Kananura, Catherine Birabwa, Felix Kizito, Sarah Namutamba, Henry Kaula, Othman Kakaire, Peter Waiswa.

**Validation:** Rornald Muhumuza Kananura, Catherine Birabwa, Moses Kyangwa, Peter Waiswa.

**Visualization:** Jacquellyn Nambi Ssanyu.

**Writing – original draft:** Jacquellyn Nambi Ssanyu.

**Writing – review & editing:** Rornald Muhumuza Kananura, Catherine Birabwa, Felix Kizito, Sarah Namutamba, Dorothy Akongo, Elizabeth Namara, Moses Kyangwa, Henry Kaula, Doreen Nakimuli, Andrew Magunda, Othman Kakaire, Peter Waiswa.

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
