## [Decision Letter · Decision Letter 0]

29 May 2023

PGPH-D-23-00486

How a co-design process led to more contextually relevant family planning interventions in emerging urban settings in Eastern Uganda

Dear Dr. Ssanyu,

Thank you for submitting your manuscript to PLOS Global Public Health. After careful consideration, we feel that it has merit but does not fully meet PLOS Global Public Health’s publication criteria as it currently stands. Therefore, we invite you to submit a revised version of the manuscript that addresses the points raised during the review process.

I invite you to address the points raised by our Reviewer 4. All the insights from reviewer 4 is very useful and needs careful attention by the authors. Kindly respond to the question about the suitability of this article to be published in PLOS-GPH. How do the authors see this paper a original research article.

We look forward to receiving your revised manuscript.

Kind regards,

Muthusamy Sivakami

Academic Editor

Journal Requirements:

1. You indicated that you had ethical approval for your study. In your Methods section, please ensure you have also stated whether you obtained consent from parents or guardians of the minors included in the study or whether the research ethics committee or IRB specifically waived the need for their consent.

2.We have noticed that you have uploaded Supporting Information files, but you have not included a list of legends. Please add a full list of legends for your Supporting Information files after the references list. 

Additional Editor Comments (if provided):

I invite you to address the point raised by our Reviewer 4. All the insights from reviewer 4 is very useful and needs careful attention by the authors. Kindly respond to the question about the suitability of this article to be published in PLOS-GPH. How do the authors see this paper a original research article.

Reviewers' comments:

Reviewer's Responses to Questions

**Comments to the Author**

1. Does this manuscript meet PLOS Global Public Health’s publication criteria? Is the manuscript technically sound, and do the data support the conclusions? The manuscript must describe methodologically and ethically rigorous research with conclusions that are appropriately drawn based on the data presented.

Reviewer #1: Yes

Reviewer #2: Yes

Reviewer #3: Yes

Reviewer #4: No

2. Has the statistical analysis been performed appropriately and rigorously?

Reviewer #1: N/A

Reviewer #2: Yes

Reviewer #3: N/A

Reviewer #4: N/A

3. Have the authors made all data underlying the findings in their manuscript fully available (please refer to the Data Availability Statement at the start of the manuscript PDF file)?

Reviewer #1: Yes

Reviewer #2: Yes

Reviewer #3: No

Reviewer #4: No

4. Is the manuscript presented in an intelligible fashion and written in standard English?

Reviewer #1: Yes

Reviewer #2: Yes

Reviewer #3: Yes

Reviewer #4: Yes

5. Review Comments to the Author

Reviewer #1: This is an exemplary co-design approach for other projects that are being implemented in developing countries to design and develop context-specific interventions on FP programs. As this HCD is an emerging approach to most developing countries’ projects, this article will have a significant positive contribution to improving the practices of project/program design techniques for different related initiatives. The project team is expected for implementing the identified interventions and will share the outcomes of the intervention with the global communities at the end of the project.

Reviewer #2: The analysis presented focuses on solution finding methodology described in through manner. Study design is well prepared and research was carried out with lose attention to the practices on the ground. A few limitations of the study which possibly exist should be reflected in the design or methodology in order to have better picture for the replicability of the study results.

Reviewer #3: Thank you for the submission.

The paper is interesting to read. This is a well-written manuscript on an important topic. The methodology is scientifically sound and rigorous. Layout of results is incredible.

Reviewer #4: The authors describe in detail the different stages of the co-design process leading to contextually relevant family planning interventions. They claim this process will lead to more relevant interventions for specific urban sub-populations.

Because the paper does not describe a research or research methods it cannot be considered for publication in PLOS Global Public Health, the first publication criterion being: ''PLOS Global Public Health is designed to communicate original research and research methods. We welcome public health research that reaches across disciplines and regional boundaries to address the biggest health challenges and inequities facing our society today.''

The authors could submit the paper to another journal, because the detailed description of the co-design process and the lessons drawn are useful for organisations developing FP interventions.

The following comments and suggestions could improve the paper (when submitted to another journal).

In the introduction the authors argue that it is important to address the unmet need for FP for different sub-populations in urban areas, in particular the poor and other vulnerable populations, because there are intra-urban inequalities. (line 93-101). However, in the description of the process, there is no referral to different sub-populations, except by gender and age-group. Following the introduction one would expect address of socio-economic differences.

Stage one of the co-design process is a mixed-methods study to understand the problem. The authors describe (203-204) that "data collected from the survey and qualitative interviews were analysed by the research team and discussed into actionable problem statements of focus during ideation". A summary of these findings by sub-population could be presented and reference made to the study report (just the study protocol is in the reference list (#31). The data should be made available.

The authors could explain how the participants for the co-design workshop were selected and which sub-populations the 13 community members represented (Line 212). A table with background of the 52 (total count not 56) workshop participants would be helpful. The same applies to the selection of participants of the test phase (line 254).

It would be interesting to present how the different groups of participants reacted to the key-insights from the study – were these new/surprising to them? (Table 2, line 302)

Table 3 (Line 308): Add to Design challenge # in first column the descriptions: Demand challenges, Supply challenges, Governance challenges (308)

Most likely, within the community participant group there have been power imbalances, by gender, age, education, etc.). Could you elaborate what you observed in the workshop? You say in conclusion: (612-615) ''For the success of the process, it is important to pay attention to participant characteristics, including the differences in education levels and position in the community, and how these could influence group dynamics and the quality of the discussions and to put in place measures to minimise the effects of this on the process.''

The practical suggestions for interventions by the workshop participants in Supporting file 1 could be more explored and commented on considering feasibility and being innovative and unexpected.

In discussion and limitations be critical of:

• Whether the small number of participants represent the communities and health workers and policy makers in the two project areas.

• In-group power differences that may have influenced the co-design process: by gender, age.

Present study findings (of phase one) on the gaps in knowledge and skills of health workers. The interventions are heavy on provider training. (Line 428) Explore reasons why they do not disclose all side effects (Line 523)– this may be intentional and not because of gaps in knowledge or skills.

In discussion: Highlight the proposed unexpected and new interventions that are probably especially effective for urban areas (compared to rural areas).

It would be interesting to see (in future) the research paper on the effects of the interventions that were co-designed. (see line 615-19) ''The intervention model prioritized for implementation will undergo further adaptations, in line with the human-centred design process, and rigorous testing to assess its effect on the uptake of voluntary FP and the reduction of unmet need for FP in emergent urban settings in Eastern Uganda.''

6. PLOS authors have the option to publish the peer review history of their article (what does this mean?). If published, this will include your full peer review and any attached files.

**Do you want your identity to be public for this peer review?** For information about this choice, including consent withdrawal, please see our Privacy Policy.

Reviewer #1: **Yes: **Samuel Muluye Welelaw

Reviewer #2: **Yes: **Dr. Kemal Goshliyev

Reviewer #3: No

Reviewer #4: No

---

## [Editor Report · Decision Letter 1]

1 Sep 2023

How a co-design process led to more contextually relevant family planning interventions in emerging urban settings in Eastern Uganda

PGPH-D-23-00486R1

Dear Ms Ssanyu,

We are pleased to inform you that your manuscript 'How a co-design process led to more contextually relevant family planning interventions in emerging urban settings in Eastern Uganda' has been provisionally accepted for publication in PLOS Global Public Health.

Best regards,

Muthusamy Sivakami

Academic Editor